# Can an Experience with No Car Use Change Future Mode Choice Behavior?

**Matus Sucha \*, Lucie Viktorova and Ralf Risser**

Department of Psychology, Faculty of Arts, Palacký University in Olomouc, Olomouc 771 80, Czech Republic
\* Correspondence: matus.sucha@upol.cz

**Abstract:** In order to determine whether an experimentally induced experience has the potential to change future travel mode choice, we recruited 10 families living in a middle-sized city who used a car at least four times a week, and made them stop using the car for one month. Each adult family member kept a travel diary and interviews were conducted prior to the experiment, after one month without a car, and then three months and one year after the experiment ended. The results suggest that the participants' attitudes towards different transportation modes did not change during the period of the study, but their actual travel behavior did. In this respect, several factors were identified that influence travel mode choice, once the participants are made aware of the decision process and break the habit of car use.

**Keywords:** car use; travel mode choice; traffic psychology; road user behavior; sustainable traffic modes

---

## 1. Introduction

Automobile dependency consists of high levels of per capita automobile travel, automobile-oriented land use patterns, and limited transport alternatives [1]. Automobile dependency has many impacts on consumers, society, and the economy. It increases mobility and the convenience of motorists. At the same time, it increases consumers' transportation costs and resource consumption, requires significant financial and land resources for roads and parking facilities, and it increases traffic congestion, roadway risk, and environmental impacts. It reduces the viability of other travel modes and leads to more dispersed land use and to mobility-intensive economic patterns that require more vehicle travel for access, thus perpetuating economic, environmental, and safety problems [2]. Both economic theory and empirical evidence indicate that excessive automobile dependency reduces economic development. Several current market distortions result in automobile dependency beyond what is economically optimal, while policies that encourage more efficient transportation and land use patterns could provide economic benefits [3].

In times of global warming, there is much talk about the contribution of private car use to this problem (Ridlington et al. [4], among many others). In 2010, a NASA study declared that automobiles were officially the largest net contributor of climate change pollution in the world. Studies mainly suggest producing more eco-friendly fuel, improving motors, building smaller cars that consume less fuel, etc. [5,6]. We adhere to the opinion that measures need to be taken but believe that technical solutions will absolutely not suffice. In our opinion it will be necessary to apply psychological work in order to achieve a substantial change. Hundreds of millions of citizens need to change their mode choice and shift from individual car use to more sustainable traffic modes, such as public transport, walking, or cycling, wherever this is reasonable and feasible. This will not be possible without their co-operation. In order to achieve such co-operation, we need to use psychological measures.

It seems that both policy makers and researchers find it difficult to address citizens with a request to use a car less, with some exceptions, such as Bamberg et al. [7] or Pike et al. [8]. The authors of these articles consider that addressing citizens and trying to achieve different mode use to reduce the use of the car should be one of the major goals of both transport research and transport policy, but they do not give any detailed hints as to what to do and how to do it [9]. Related research is scarce both before and after these publications. In fact, achieving a change of mode choice appears to be a difficult task, considering that we all seem to depend on car use. However, concerning Europe, it is known that ~50% of all car trips are shorter than 5 km and ~30% are shorter than 3 km. In the United States of America, according to the Federal Highway Administration [10], about 10 billion miles p.a. are covered in the frame of car trips of under a mile. When there is no attractive public transport, such distances can be covered by cycling and walking. Efficient public transport allows much greater distances to be covered by combining walking and cycling with public transport.

A possible trend in this direction might be represented by the fact that young people are no longer getting driver's licenses to such a great extent as hitherto. According to a study by Michael Sivak and Brandon Schoettle at the University of Michigan Transportation Research Institute, the percentage of people with a driver's license in the USA decreased between 2008 and 2010 across all age groups. For people aged 16 to 44, that percentage has even been decreasing steadily since 1983 [11]. The top three reasons for not obtaining driving licenses were: "too busy or not enough time to get a driver's license" (37%), "owning and maintaining a vehicle is too expensive" (32%), and "able to get transportation from others" (31%). Another reason that comes to mind is that more people are living in cities and using public transport. But in the survey, only 17% said their reason for not having a license was that they preferred public transport.

For Germany, Kuhnimhof et al. [12] showed that since the turn of the millennium, car use has not been increasing any more among young adults, especially as far as males are concerned, while the use of active modes (walking and cycling) and of public transport is increasing slowly but steadily.

The notion that there might be a need to reduce car use is not as new as it might seem. There were already thoughts, arguments, and discussions in that direction by the end of the eighties and beginning of the nineties of the last century. At the beginning of the 1990s, the European Union (EU) started financing projects that dealt with questions of mode choice. This was a consequence of the first critical reports on the car as a transportation mode: that beside its undeniably big advantages, car use also seemed to cause problems that were starting to become obvious—the sealing of the ground, increasing crowding of cities, air pollution, noise problems, health problems caused by both air and noise pollution, and the fact that citizens of all ages were reducing their levels of physical activity by using cars more and more, even over very short distances. In the literature, the disconnection of local communities as a result of increasing car use was taken up as early as 1974 by Kasarda and Janowitz [13], and similar arguments were brought forward in 2015 by Stephen Moss [14]. The fact that car use regularly contributes to a decline in local economies was analyzed and discussed by, among others, Handy and Clifton [15]. Maibach et al. [16] estimated the societal costs of increasing obesity and cardiovascular diseases and the consequences of air pollution, to which car dependency contributes. Additionally, the economic burdens imposed by the emission of greenhouse gases on urban sprawl and decay and the detrimental effects on active mobility modes—which, on many occasions, e.g., for short distances, could be used instead of the car—were included in their calculations. In 2013, Flade [17] summarized much of what had been dealt with up to then concerning mobility behavior, how people have become dependent on the car as mobility modes, and what obstacles there are to achieving change.

However, all the studies mentioned so far focused on the identification of the problems connected to car use and on underlining that things should change, with formulations in the conclusions of their analysis papers, but concrete recommendations as to how to replace car use with other modes were mostly lacking. There is an implicit understanding that if car use is to be reduced, the cooperation of citizens is needed. It is logical that there will be no change if individuals do not change their behavior. But how can that cooperation be achieved?

The answer is that human behavior, motives, and attitudes have to be taken into account, not only in the frame of analyses, but also in connection with the development of measures. Behavioral change can certainly not be achieved without including sciences that deal with human behavior, in the first place, psychology. Psychology is the discipline that, according to most definitions, deals with individual human behavior, its understanding, and ways to influence it. Among others, a research briefing sheet of the Center for Transport and Society [18] of the faculty of the built environment in Bristol summarized in detail what role psychology should play in connection with our mobility. Schlag et al. [19] displayed the basis on which mode use can be understood and steered.

In the context of this article, the focus is on understanding that "car consumption is never simply about rational economic choices, but is as much about aesthetic, emotional and sensory responses to driving, as well as patterns of kinship, sociability, habitation and work" [20]. If one wants to replace car use with other mobility modes, one needs to see to it that complicated individual and social functions are taken over by those other modes.

Replacing car trips with active modes, how it could be achieved, and what positive effect it would have were already discussed by Holtz Kay as early as 1998 [21]. Among others, the EU projects WALCYNG (Walking and cycling instead of short car trips [22] and ADONIS (Analysis and development of new insight into substitution of short car trips by cycling and walking). Dijkstra et al. [23] were implemented at the end of the last century; they summarized the problems with car use identified so far and dealt with the question of how to replace (at least short) car trips with walking and cycling. While these projects and (only a few) other research projects at that time accepted the idea that psychology had to be applied in order to achieve changes in citizens' mobility behavior, not much use was made of this idea in practice. For instance, the concept of providing incentives was mentioned in theoretical texts. WALCYNG [22] referred to one successful step taken by a company in Vorarlberg (Austria), where the owners succeeded in substantially reducing car use by their employees by providing incentives. This allowed them to reduce the size of their parking facilities and use the space for storage instead. However, the use of incentive programs in practice at that time and also in the years to follow was near to zero, as were research projects dealing with the question of how to make people change their mobility behavior.

What is so interesting about incentives? They are a way to make people try out something that they would not do without them, thereby touching all the aspects mentioned by Sheller [20]. As we will see below, trying out and experiencing things is a key issue in connection with attempts to change behavior. In an ideal world, citizens would like to make use of alternatives instead of going by car; they would not have to be convinced to do so. The concept of "renouncing" something by not using the car would become obsolete. Instead, a change would be perceived as a gain. Theoretically, this could be achieved if road users could feel the advantages of different mobility habits from those of today, instead of just intellectually understanding the arguments that a change of mobility habits away from the car would be wise. The problem that this argument displays is, in fact, that one can only feel the advantages of certain behavior if one performs it.

In marketing practice, this problem is tackled with the help of incentives [24]: encourage people to try out a certain "thing"—a behavior, a product, a recipe—and by trying it out, they will find that this "thing" is attractive, efficient, and effective; in other words, it provides positive feelings. In psychological terms, extrinsic motivation will, it is hoped, be converted into intrinsic motivation. People will like or start liking the "thing".

The risk is, of course, that someone who offers incentives is wrong when he/she believes that what is offered and then tried out will be perceived positively by the addressees. That is why good preparation is needed. One needs to understand how the addressees might look at what is being offered. Do they agree that what the experimenter thinks feels positive really does? What aspects might be experienced as negative? In the beginning, these questions can only be dealt with in an exploratory and qualitative way. Understanding motives is not really possible with the help of a quantitative approach, as this would require questions leading to answering—"ticking"—lists that contain all the

relevant possibilities to explain possible predilections, barriers, prejudices, assumptions, etc. Such a list, produced by the research leaders themselves, would of course run the risk of being heavily biased if the persons who were interviewed had aspects in mind that the researchers did not think of or not consider relevant. In the frame of such an approach, one would have people try the "thing" out on a smaller scale, find out what worked and what did not, summarize the results, and carry out the next study with improved and more detailed knowledge, and so on. This is exactly what Burwitz, Koch, and Krämer-Badoni [25] started in Bremen, Germany, in the 1990s. They conducted a qualitative experimental study for which they recruited seven families living all over the city and convinced them to stop using their car for one month. Each family member kept a travel diary and interviews were conducted at the beginning and at the end of the experiment. Their results suggest that if people are made aware of their travel mode choice, it is possible to change their habitual behavior. They also pointed out various motives that people might have for giving up car use, as well as the difficulties and positive experiences they might face when doing so and made suggestions to the city council pertaining to suitable ways of promoting car use reduction and changes in public transport.

In this way, a knowledge base could be established stepwise. Unfortunately, such research is hardly ever financed nor published [26], and therefore the state of the art is poor and a knowledge base is lacking. The call for "representative results" and for quantitative outcomes reflected in figures has become a strong tradition that makes it difficult to carry out qualitative and exploratory studies that serve to provide a better understanding of the feelings and motives of people. However, such an understanding is necessary in order to have a chance to achieve behavioral change. The present study focuses on the exploration and description of the psychological factors which influence mode choice, especially those which make the choice of any mode attractive. The implications of this work can be used in the promotion of sustainable traffic modes.

## 2. Theoretical Considerations

In this section, we present theories which are relevant for the traffic mode choice and whose central element is human behavior or, more specifically, the factors which influence human behavior. Psychological and, more specifically, sociological psychological approaches are presented. Three main groups of theories have to be considered: theories representing behavior as a reasoned action, theories dealing with habits, and theories dealing with the process of influencing and changing behavior. Even though there are also relevant sociological theories, for example, the mobility biographies approach or social practice theory, we do not discuss them, as we follow a psychological approach in our research. The central element of our research and theoretical considerations is the role of habits.

### 2.1. Psychological Aspects of Traffic Mode Choice

As a mode change "away from the car" is the implicit goal, adherence to the car needs to be discussed first. In today's world, many of us are socialized to use a car. Our parents did so and we do it too. When we grow up and reach the age when a driving license can be obtained, it is usual to do so and thereafter to get a car. When one is looking for a job, one of the common requirements is a driving license. Other factors which boost car use are urban architecture and infrastructure, which are very car-oriented, often to the disadvantage of other modes of transport [27]. Even if young people decided earlier not to use (and own) a car, they usually start using one, nevertheless, once they start family life—often when a child is born. Our "world" tells us that it is normal to use a car and that it is the first choice. It is only if something goes wrong (no money, the car has broken down, suspension of our driver's license) that we think of other possibilities.

In psychological terms, we develop—or have already developed—a habit. We do not think about our decisions and their consequences; we usually do not consider which mode of transport we will choose "today" and then decide. Instead, most of us use a car habitually [28]. As is very well known from psychology, changing habitual behavior is rather complicated. Often, it is not rationally founded (any more), but just performed automatically without any reflection. According to Prochaska and

DiClemente [29], we are in a pre-contemplation stage, where we see no hint at all that it could make sense to change something in our behavior.

One of the problems when changing habitual behavior is what is called the "problem of the starting point" or "endless circle". This means that we need a "disturbing" experience, something that gives us a hint or a feeling that we might need to change certain behavior. At the same time, we need to do more than only hear or read that a new behavior is positive. We also need to feel that a change of behavior is good and enjoyable in order to understand it fully. But this is only possible if we practice certain behavior. Thus, as we can see, we need something as a starting point that makes us try out a new behavior. The problem is, however, that our habits and routines do not easily provide such a starting point—a vicious circle. In the study referred to in this paper, we created a situation where the test participants had to change their habitual behavior for a short time and we checked whether this could be one way to interrupt this vicious circle. This has not been tried very frequently before, one of the few exceptions being the precursor of this study [25]. We are of the opinion that more attempts of this type are needed.

## 2.2. Psychological Traffic Mode Choice Theories

Traffic mode choice theories which are based on psychological principles focus on the factors which influence the decision-making process and behavior. Generally, two groups of factors are discussed. One group is constituted by external factors, or situational factors, such as social influences or the availability and attractiveness of the modes (i.e., the distances to the public transport stops, etc.). The other group consists of internal factors, which define how a person perceives reality (i.e., cognitive and attribution processes), such as the influence of past experience, habits, and attitudes. Social psychology provides theoretical frameworks to take into explicit account the effect of different psychological and contextual conditions upon the decision-making process [30].

The theory of planned behavior (TPB) is widely used as a behavioral frame of reference [7]. The TPB explains that human behavior is affected via intention and perceived behavioral control. The intention is assumed to be affected by three factors, including the attitude towards the behavior, subjective norms, and perceived behavioral control, while behavior depends on the motivation to comply with norms. TPB proposes that behavior can be explained by behavioral intention, which in turn is influenced by attitudes towards the behavior, subjective norms (beliefs about other people's expectations), and perceived behavioral control (freedom of choice and to what extent the behavior will be difficult to perform). The results of several studies [31,32] show that this theory explains transport mode choice very well. Behavioral intention explains between 69 and 82% of the variation in transport mode choice. Together, attitude, subjective norms, and perceived behavioral control explain between 49 and 72% of the variation in behavioral intention [33]. But in our context, we assume that this explanation is ex-post. A habit is there already, attitudes and subjective norms are comfortably adapted to this habit, and, of course, behavioral control is high. There is nothing easier to perform than habitual behavior.

Habitual behavior is also the main predictor to explain travel behavior. Habits are characterized by three features: automaticity (we do not reconsider our decisions in every situation), situation constancy (in unchanged situational settings, we tend to take the same decisions), and functionality (if the behavior that is performed is functional, or, so to say, "good enough", we tend to maintain it; Verplanken and Aarts [34]. Accordingly, the studies by Chen et al. [35], Chen and Lai [36], and Eriksson et al. [37] show that habit affects mode choice intention and behavior.

According to the theory of habits, human social behavior is only partly reasonable. People's beliefs are unfounded or biased, and their attitudes, subjective norms, and perceptions of behavioral control have a very strong influence on their behavior. This view suggests that human behavior can be automatic or habitual. The frequency of past behavior is an indicator of the strength of a habit, and it can be used as an independent predictor of later action.

The theories about habits propose that repetition of any behavior may lead to this behavior becoming automatized, performed without noticeable effort or attention. The repetition of the behavior is dependent on the person's experience with this behavior. If it functions well, does not lead to negative consequences, and fulfils its functions, a habit is established. There are reasons why we repeat a behavior, such as advantages that we perceive, and that maybe society even appreciated at a certain (earlier) point in time. On the basis of the previous behavior, a habit is developed and then behavior starts to be performed without further reflection. Disadvantages, including those developing in the course of time, both for the individual and for society, may thus be neglected or not perceived. Frequent use of a specific mode of transport on a repeated journey, having become a habit, may cause the same mode of transport to be chosen on future repetitions of the journey without the traveler going through any thorough decision-making process, or without even being conscious that there is, or could be, a choice. Cognitive science describes this as Luchin's effect, which refers to a person's predisposition to solve a given problem in a specific manner even though better or more appropriate methods of solving the problem exist. This effect describes the fact that people frequently make use of habitual solutions or choices even in situations where there are clearly conceivable better ones [38]. The results of research [33] showed that a habit of using the same mode of transport for several different journeys reduced the amount of information being used in the decision process.

The theory of interpersonal behavior (TIB) by Triandis [39], further elaborated by Galdames et al. [40], takes habit into account as one of the influencing factors as well. This theory suggests that observed behavior corresponds to an intention that is mediated by habit (frequency of use) and facilitating conditions (contextual situation). In our case, the theory of interpersonal behavior (TIB) suggests that mode choices depend on individual attitudes towards available modes and social influences, habits, and facilitating conditions [40].

Further, we present theories which are widely discussed in the literature and deal with habits, habitual behavior, and, more specifically, with key factors influencing or changing these habits and consequently behavior. This is especially important because in our research, we use incentives to influence behavior and, possibly, to change habits. These theories are: the transtheoretical model (TTM) by Prochaska and DiClemente [29], the model of normative decision making (NDM) by Schwartz and Howard [41], and the comprehensive action determination model (CADM) by Klockner and Friedrichsmeier [42]. In all of them, the determining effect of habits—as a kind of state of pre-contemplation—plays an important role.

In connection with the perspective taken by the TTM, it is important to understand the relationship between reading or hearing about the advantages of a certain (new) behavior and the disadvantages of a certain (old and unwanted) behavior on the one hand, and, on the other hand, becoming motivated by such information to change something. The "new" behavior needs to have more motivational weight, which could be represented according to Figure 1 below [43], which does not deal with mode choice but with speeding as a habit:

The benefits of certain behavior need to be perceived. Of course, the authors of this paper, as researchers working in the field of mobility, know a lot of benefits of a mode choice that is independent from car use. The problem is, however, that such knowledge is quite abstract and does not necessarily motivate people to think about a possible change in their behavior. Marketing literature, on the other hand, shows that colorful descriptions by peers concerning certain types of new behavior are potentially much more motivating [44]. One of the main thoughts that shaped our approach was to make use of this principle, in addition to the idea of providing incentives. A qualitative procedure, striving for saturation with respect to reported experiences that one could have when trying out car-free mobility for a certain time, appeared to us to be appropriate in this respect. The "colorful" descriptions would be found in the reports of the participants in our test, we assumed.

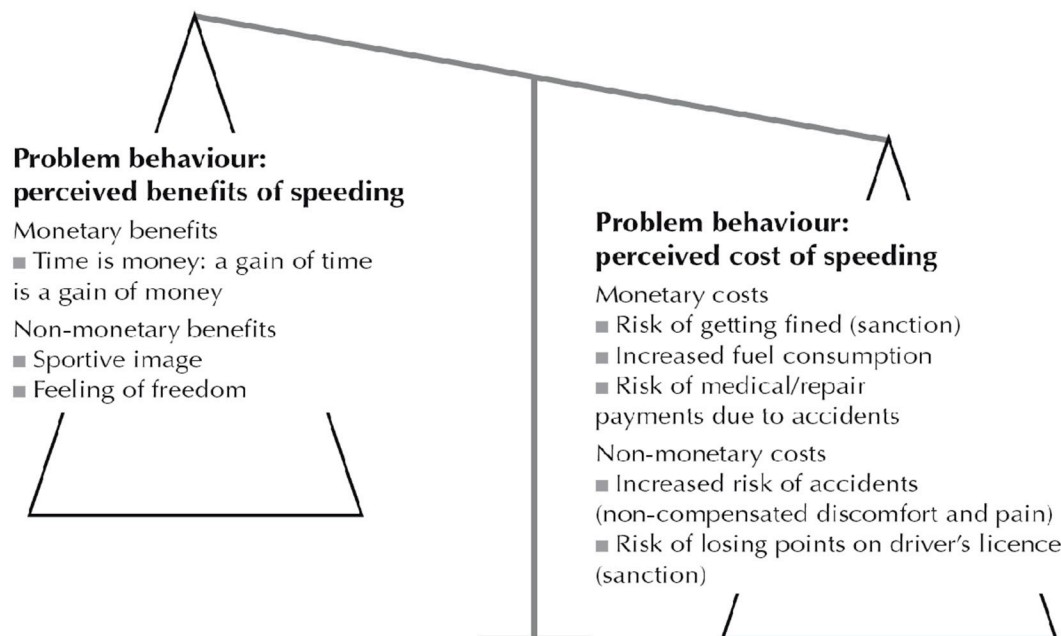

**Figure 1.** The "motivation balance" [43].

On the basis of the background and relevant theories presented above, we formulated our hypothesis and expectations (below). We wanted to make people interrupt the vicious circle of habitual behavior by inviting them to take part in our experiment; thus, we led them—we hoped—to experience the ease and good quality of alternative modes (which, we hoped, should affect their attitudes in a wished-for way), and we expected to extract from this process "colorful descriptions" that would be convincing and motivating. One of the questions to be answered was whether the mode-choice behavior of the persons involved in the experiment would change. Other results would consist of ideas and notions for improvement provided by the participants, to be forwarded to planners and decision makers.

## 3. Methodology

### 3.1. Aim of the Study

The study presented in this paper was inspired by an exploratory research study that was conducted at the beginning of the nineties of the last century and that dealt with the question of how life without a car was perceived by test persons or test families [25,45]. In this research, test families were incited to try life without a car for one month within the frame of a scientific study. The repetition of this study in the Czech Republic built on the results of this earlier study in order to, it was hoped, further improve understanding of both the perceived disadvantages and advantages of a life without a car. Our main research questions were:

(1). What advantages and disadvantages are perceived (or experienced) during one month without a car?

(2). To what extent can a month without a car influence a change in travel behavior?

A further aim was to communicate positive experiences to the general public and make suggestions to the authorities and stakeholders as to what should/could be improved with respect to alternative mobility options.

*3.2. Research Sample and Recruitment*

As we wanted to replicate the study by Burwitz, Koch, and Krämer-Badoni [25] as closely as possible, the aim was to recruit 10 families living in the city of Olomouc (Czech Republic) or up to 50 km away, who used a car at least four times a week, and make them stop using the car for one month. In May and June 2017, emails were sent to university employees inviting them to participate in the research. Additionally, an ad was created and posted in the university journal and in two local newspapers (one published by the city hall, one commercial newspaper) with basic information about the research. Participants were promised 4000 CZK (around 150 EUR) per family for participation. In reaction to this, 40 emails were received. Upon further explanation being given of the details of the study and the start of the experimental period in autumn 2017 (in order to avoid an irregular travel schedule during the summer holidays), 37 people claimed to remain interested. In September 2017, further emails were sent to those who had indicated their interest, requesting them to fill out a short form pertaining to the number of family members taking part in the study (including children), their home address and contact information, and suggestions for times when the family could be contacted by a research assistant in person for the initial interviews. In total, 17 persons responded to this email and filled out the questionnaire. On the basis of the indicated number of family members and the location of the household, 10 families were selected, so that there were:

- six families with four members
- two families with three members, and
- two families with two members.

All of them were living in different parts of the city (the southern part of the city near the main train station, the city periphery, suburbs up to 5 km from the city, and a town about 20 km away). During the first two weeks of the experiment, one family of four had to be replaced as a result of their inability to arrange a meeting with the research assistant, and one family of two dropped out at their own suggestion because of a sudden medical emergency that prompted the family to use the car regularly. Both families were replaced by other families that had previously not been selected: one family with two members (a father and a daughter), and one family with a newborn child (five family members in total). Detailed information about the final set of families is presented in Table 1. We assumed that characteristics such as the distance to the workplace, supermarket, or nearest public transport, as well as the number and age of children, might influence willingness to use different transportation modes.

**Table 1.** Family characteristics.

| F. | Location | Total No. of Family Members | Adults | Children | Children's Ages | No. of Cars | No. of Bicycles | Gross Monthly Income (Family, EUR) | Average Monthly Car Costs (EUR) | Distance to Public Transport (in Metres) | Distance to Train Station (in Metres) | Distance to Supermarket (in Metres) |
|---|---|---|---|---|---|---|---|---|---|---|---|---|
| 1 | north periphery | 2 | 2 | 0 | / | 1 | 2 | 1925 | 115 | 500 | 4000 | 500 |
| 2 | northwest periphery | 4 | 2 | 2 | <10 y.o., nurseling | 1 | 3 | / | 115 | 600 | 3500 | 3000 |
| 3 | near main train station | 4 | 2 | 2 | 3 y.o. (preschool), nurseling | 2 | 7 | 1230 | 50 | 5 | 300 | 800 |
| 4 | near main train station | 5 | 2 | 3 | 11 y.o., 3 y.o. (preschool), newborn | 1 | 3 | 1270 | 80 | 5 | 300 | 800 |
| 5 | suburbs (<5 km) | 4 | 1 | 3 | 12 y.o., 9 y.o., 3 y.o. (preschool) | 1 | 3 | 1925 | / | 300 | 6000 | 5000 |
| 6 | northwest periphery | 2 | 1 | 1 | <10 y.o. | 1 | 2 | 1155 | 40 | 300 | 5000 | 500 |
| 7 | city 20 km away | 3 | 2 | 1 | <10 y.o. | 1 | 2 | 1350 | 65 | 500 | 2000 | 2000 |
| 8 | suburbs (<5 km) | 3 | 1 | 2 | both <10 y.o. | 1 | 1 | 1540 | 155 | 300 | 1000 | 3000 |
| 9 | near main train station | 4 | 2 | 2 | both <10 y.o. | 1 | 4 | 1730 | 190 | 20 | 500 | 500 |
| 10 | northwest periphery | 4 | 2 | 2 | 10 y.o., <10 y.o. | 2 | 4 | 1155 | 115 | 500 | 1000 | 700 |

Although we generally wanted children older than 10 years to participate actively (write a diary and travel log, fill out the questionnaires), none of the eligible children decided to do so (or the parents did not see this as a viable idea). Therefore, we only have data from the adults. A summary of the demographic data for the 17 individuals is presented in Table 2. In this matter, it is noteworthy that almost all our participants were university graduates of productive age (25–50) who had children (similar to the sample of Burwitz, Koch, and Krämer-Badoni) [25], so they might feel the need to have a car, as well as possibly having sufficient finances to allow them to possess one.

**Table 2.** Individual characteristics.

| Family | Gender | Age | Education | Family Status | Distance to Workplace (in Meters) | Car Use/Week | Public Transport/Week | Cycling/Week | Walking/Week | Health Status (1-Poor; 5-Excellent) |
|--------|--------|-----|-----------|---------------|-----------------------------------|--------------|------------------------|--------------|--------------|-------------------------------------|
| 1 | F | 25 | university | partnership | 1500 | 4 | 7 | 0 | 5 | 4 |
| 1 | M | 26 | university | partnership | 2000 | 4 | 2 | 5 | 2 | 4 |
| 2 | F | 35 | university | married | maternity leave | 3 | 5 | 3 | 2 | 5 |
| 2 | M | 37 | university | married | 3500 | 4 | 3 | 2 | 7 | 4 |
| 3 | F | 34 | university | married | maternity leave | N/A | N/A | 1 | 7 | 4 |
| 3 | M | 33 | university | married | 400 | 2 | 0 | 7 | 7 | 5 |
| 4 | F | 37 | high school graduate | married | maternity leave | 4 | 1 | 0 | 2 | 5 |
| 4 | M | 37 | university | married | 2000 | 4 | 2 | 2 | 5 | 4 |
| 5 | M | 37 | university | separated | 7000 | 3 | 3 | 4 | 2 | 4 |
| 6 | M | 50 | university | divorced | 3000 | 7 | 1 | 1 | 3 | 5 |
| 7 | M | 29 | high school graduate | married | 400 | 5 | 0 | 5 | 7 | N/A |
| 7 | F | 30 | university | married | 24,000 | 5 | 0 | 1 | 7 | 5 |
| 8 | F | 36 | university | widowed | works from home | 5 | 0 | 1 | 7 | 3 |
| 9 | F | 37 | university | married | 2000 | 6 | 1 | 5 | 4 | 5 |
| 9 | M | 38 | university | married | 3700 | 3 | 2 | 6 | 3 | 5 |
| 10 | M | 36 | university | married | 7000 | 5 | 0 | 3 | 1 | 5 |
| 10 | F | 33 | university | married | 7000 | 7 | 1 | 0 | 1 | 5 |

### 3.3. Study Design

A repeated measures design was chosen: at the beginning of the experiment (October/November 2017), each family member over the age of 10 completed the World Health Organisation WHO quality of life questionnaire (WHOQOL-100; the results will not be reported here) and a short questionnaire containing items regarding the frequency and attractiveness of car use (as perceived by the family members) and other transportation modes such as walking, cycling, or public transport. Then the families were interviewed about their attitudes towards car use and asked to keep a "travel diary" and a log-book (Google maps application) with daily entries on the approximate distance travelled via different modes of transport, in order to gain a better understanding of their travel habits, purposes, and travel mode choice. For the first week, they were asked to travel "as usual" and then the one-month period without car use began. The participants were discouraged from using a taxi service or regular car rides with acquaintances, but these were not strictly forbidden. After this month, the families were again interviewed about their experiences and asked to fill out questionnaires once more. A final interview and questionnaire collection took place three months after the end of the experimental period (March 2018), in order to assess the change in car use. A follow-up assessment approximately one year after the beginning of the study (November/December 2018) was also conducted, this time using only a short questionnaire on transportation mode use, in order to determine the duration of the respective changes.

### 3.4. Data Collection Methods

Use and attractiveness of different transport modes questionnaire.

For each transportation mode (car use, public transport, bicycle, and walking), the participants were asked to rate the perceived comfort, time consumption, financial costs, and overall attractiveness on a five-point Likert scale. The items were re-coded so that a higher value means a more positive rating (more comfortable, less time-consuming, less costly). Furthermore, the participants were asked to indicate how many days a week (zero to seven), on average, they used the respective mode of transport and how many trips per day, on average, they made via this transportation mode (none, up to two trips, up to four trips, more than four trips). This questionnaire served to gain an overview of the participants' attitudes and transportation mode usage at each data collection point.

### 3.4.1. Travel Diary

Apart from noting all the trips and their duration in the travel log, each day of the study (one week travelling as usual + four weeks without a car), the participants were required to answer the following questions:

- What did you have to organize differently as a result of the absence of the car?
- Why did you choose the transportation modes you chose today?
- What problems occurred while using these transportation modes? What did you see negatively?
- What did you perceive as positive while using these transportation modes?
- Did you experience something that you probably would not have experienced if you had been using a car?
- Did you regret that you could not use a car today? Why?
- What else do you perceive as memorable today?

For each entry, a minimum of 10–30 words was recommended; however, the participants did not always adhere to the given minimum word count.

### 3.4.2. Interviews

The interviews usually took place with the whole family together, so that the members could react to each other. At the initial interview, the purpose and design of the research and the conditions for taking part in it were explained to the participating families, as well as how to keep the travel log and diary. Each family was then asked about:

- their motivation to take part in the experiment and their expected experience with it
- whether they had previously tried to live without a car (and if so, additional questions about that experience were asked)
- the purposes for which they usually used the car
- what reasons they had for their transportation mode choice.

After one week of using the car as usual, problems that arose with keeping the travel diary were cleared up and the participants were instructed to start their "life without a car". Four weeks later, another interview was conducted, focusing on:

- the differences in daily routines and life in general compared to daily life when using a car
- situations for which a car was (and is) perceived as advantageous
- planned changes in the prospective life of the families regarding mode use.

The last interview took part three months after the end of the experimental period and the following aspects were assessed:

- whether the families returned to their "old habits" or kept up (some of) the new arrangements
- how they perceived their experience with life without a car now
- if they planned to continue their life without a car or to return to using it.

## 3.5. Analysis of Data

In order to determine the advantages and disadvantages perceived and experienced during one month without a car, and to communicate the positive (and negative) experiences with different modes of transport, the data from the interviews and travel diaries was analyzed. Grounded theory was used as the main approach, starting with in-vivo and open coding of the experiences and further summarizing them into categories and sub-categories, with relevant examples (see Table 3). Furthermore, on the basis of the travel diary entries, frequency tables of different positive and negative experiences and their occurrence were created for each mode of transport (car, bicycle, walking, public transport, and trains) separately, summarizing the experiences of the participants; these are presented in the Results section.

**Table 3.** Categories and sub-categories used to classify positive and negative aspects of transportation modes, with positively and negatively rated examples.

| Category | Sub-Category | Types | Example—Positive | Example—Negative |
|---|---|---|---|---|
| weather | - | - | sunny | cold, rainy |
| destination | infrastructure | – availability of bicycle paths <br> – road condition <br> – parking spotspavements | bicycle paths and pavements are available; with a bicycle—no need to look for parking spots | bad road condition, no bicycle paths or pavements, no parking spots |
| | distance | - | close | far away |
| traffic conditions | time | – speed <br> – delays <br> – time schedule | travelling on a bicycle is quick | bus available only every 30 min; long train delays; waiting for public transport; walking takes time; getting up earlier |
| | available connections | – availability "at all" <br> – planning ahead/flexibility <br> – finances | good connections; maintaining a bike is cheaper than maintaining a car; one can manage more trips in a day with a car; one does not have to rely on others/time schedule | no train/bus connection to the destination; need to buy tickets in advance; need to plan the whole day more carefully; expensive train tickets |
| | comfort | – external conditions <br> – psychological comfort | wifi and toilets in trains; relax, no need to focus on driving, possibility of reading, possibility of drinking alcohol | traffic congestion, crowded public transport, no place to sit; stress of managing everything, tiring |
| | social aspect | – behavior of others <br> – family and children <br> – friends and relatives | nice social interaction on trains; possibility of playing with children on trains; children learn to use public transport on their own; riding more with friends | recklessness of others (drivers/public transport), no privacy on public transport; walking makes children get tired, need to look after small children; not visiting relatives who live far away that often |
| | safety | – day time <br> – overall | - | not safe to travel alone at night by public transport or walking; car accidents; not safe to transport children in a carriage behind the bicycle |
| baggage | - | - | one can transport everything with a car, including building material | bigger shopping not possible on foot/with a bike/on public transport; |
| environment | surroundings | - | enjoying the countryside, exploring the neighborhood | - |
| | ecology | - | good conscience about not polluting the air | pollution |
| health | - | - | physical exercise with walking and cycling—good feeling | illness or sudden health issues triggered the need to use a car |

As for the second goal, to answer the question of the extent to which a month without a car can influence a change in travel behavior, a mixed-methods approach was used. First, data from the short transportation modes questionnaire (at baseline, one month-, three months-, one year after) was examined. To assess the use of different transportation modes, only the "days per week the mode was used" were analyzed. The answering options of the "trips per day" item (i.e., "none", "up to

two trips", "up to four trips", "more than four trips") turned out to be unsuitable for the participants, as they had trouble indicating correctly the average number of times they used the respective mode of transport each day. For this estimate, the log-book entries proved to be more accurate. But even there, some participants tended to "split" their trips according to every stop they made (e.g., they left work at 4 p.m. in their car, stopped in the city center for 15 min to shop, and then continued with the car to their home, counting each part of the journey as a separate trip), whereas others only made "rough" entries (e.g., "trip from work—1 h"). Therefore, we decided to focus more on the car use/other transportation mode use purposes, as reported in the travel diaries and in the interviews, and less on the "exact" number of times per day the respective mode of transport was used. Still, to give at least some rough estimate of the transportation mode usage at each stage of the study, the differences in "days per week usage" of the given transportation modes between the data collection times were analyzed using parametric tests (repeated measures ANOVA with Bonferroni post-hoc tests). Because the data was incomplete for two participants (unavailability at one of the times of measurement), only 15 complete datasets remained and were analyzed.

In a similar way, the change in attitudes towards different transportation modes between the respective data collection points was examined, using the short transportation modes questionnaire. Because of the data distribution, non-parametric tests for repeated measures (Kendall's coefficient of concordance) were used to examine the differences on the Likert-scale items (attitudes).

In this way, families and/or individual participants were identified who seemed to change their travel behavior patterns (and those who did not). Next, we turned back to the data from the interviews and confronted the quantitative findings with the statements of the respondents regarding their change in travel behavior at different time points. This was done to further verify the identification of respondents whose travel behavior seemed to have changed. Finally, we tried to identify common characteristics of the respondents whose travel behavior seemed to have changed as a result of their participation in the experiment and of those who went back to using a car after the experimental period ended. This was done as in the axial coding phase of grounded theory, when demographic characteristics and categories that were identified in the interviews and from the travel diaries were connected to each other. The categories used in this phase of data analysis focused mainly on the car use purposes (e.g., "shopping", "work", "school", "leisure time activities", "visiting relatives", and "other trips") and reasons for using a specific mode of transport (e.g., "comfort", "time", "flexibility"—which considers both time and comfort—"weather", "finances", "safety", "destination", and "baggage").

## 4. Results

*4.1. Perceived Advantages, Disadvantages, and Experience with Different Travel Modes—Data from Interviews and Travel Diaries*

Among the factors that were discussed most by the participants regarding their travel mode choice and experience were time, psychological comfort, baggage, flexibility, and weather, followed by infrastructure in general, parking spots specifically, health, and children. Less often, finances and traffic conditions were mentioned, as well as the behavior of others (in traffic), distance, family, and appreciation of the surroundings. Also, the (in) ability to drink alcohol, the perceived safety, social interactions, ecology, or external conditions of the transportation mode related to comfort were mentioned several times.

Time referred to different aspects, e.g., "saving time" when the transportation mode was perceived as "quick" (or "quicker" compared to a different one; mostly car vs. public transport and bike vs. walking), but other issues, such as the need to get up earlier, waiting for a connection, or delays, were mentioned in this respect as well. Psychological comfort encompassed the overall perceived comfort of the mode of transport (usually lower for walking and public transport and higher for a car), but also the opportunity to relax or maybe even read a book on the way, and the good feeling of "not using a car this time". On the other hand, stress was a part of this category, too, and it was perceived both while using a car (mostly in connection with finding a parking spot and returning back

in time so that one does not have to pay a higher parking fee) as well as in other modes of transport (e.g., catching a train or other time-related issues). Another stressful factor was the current traffic (i.e., a lot of cars), especially in connection with the infrastructure—the availability of pavements and crossings, roads for cyclists, and the state of all the roads, but also the infrastructure of train stations (e.g., a place to sit that is shielded from weather conditions) and the state of the vehicles (mostly trains), i.e., external conditions, in general. In this matter, people appreciated when there was an "other-than-car user-friendly" infrastructure available and complained if there was not.

The respondents also seemed to choose the respective mode of transport on the basis of the distance they had to travel, e.g., appreciating walking for shorter distances and other modes for longer distances (with the exception of a bike for inter-city travel). In this matter, weather plays a role, too, in that people seem to be more willing to walk or use a bike if the weather is good but turn to other modes of transport if it is cold, rainy, or dark. They often complain, then, about the temperature conditions in public transport or having to think about how one should dress to avoid discomfort. The amount and type of baggage seems to be another reason why people prefer different modes of transport, especially when travelling with children. When people need to transport a special kind of baggage, e.g., bricks or other building material, using a car is almost unavoidable; but even if one would like to "just" do a larger shopping or take more suitcases (e.g., for the children's stuff), people prefer using a car.

As for children as a category on its own, the respondents saw both positive as well as negative aspects of using different transportation modes. Some were happy that their (10+ y.o.) children learned to use public transport on their own or that they, as parents, could play with their children during the ride (in a train), but some noticed that the children get tired easily, that they have to pay more attention to their safety, or that the children have a lot of leisure time activities to which they need to be transported, and various modes proved to differ in their suitability for this purpose (e.g., a car was appreciated more than riding a bike). This also had to do with the overall perceived safety of the mode of transport, as well as with its perceived flexibility. Some respondents appreciated not being dependent upon the car and the state of the traffic, but in other cases, they did not like being dependent on public transport and train schedules either, and all of them mentioned the need to plan ahead as something they had to learn and that was not that easy.

Compared to the factors mentioned above, financial costs seem to be rather secondary, with people appreciating cycling and walking as less costly, but still preferring to use a car for the reasons mentioned above—or when they realized that a train ticket might be more expensive than using a car for a family trip (for more than three family members). Similarly, ecology was also mentioned only rarely—even less than the appreciation of the possibility of drinking alcohol if not using a car.

Overall, depending on the context and traffic mode, all of these factors were mostly perceived as being both positive and negative. The only exceptions were the following aspects, which were either mentioned in a rather positive way, or not mentioned at all:

- health in the sense of "doing something for one's health" (walking, cycling), but also in the way that if one (or one's children) is sick, some transportation modes (e.g., a car) are more useful than others—or as one participant put it, "a car is a great helper if you have health issues or if you need to get your child to the hospital quickly";
- although the behavior of others, such as being rude, loud, sick, or just overcrowded, was mostly viewed as a negative aspect of using public transport or a train, social interactions with strangers or friends, on the other hand, were rated positively, and were noted in all transportation modes except when cycling;
- interestingly, spending quality time with the family, such as playing games and talking to each other, was not mentioned in the case of using a car, but it was perceived as a positive aspect of all other modes of transport;

- and finally, observing or discovering new surroundings, and positive feelings connected to the experience of nature, were also mentioned in connection with all other transportation modes except a car.

Table 4 summarizes the occurrence of positive and negative aspects with the various modes of transport under study, as reported by the participants in their travel diaries.

**Table 4.** Categories occurring as positive and negative aspects of different modes of transport.

| | BIKE | | WALKING | | PUBLIC TRANSPORT | | TRAIN | | CAR | | TOTAL | |
|---|---|---|---|---|---|---|---|---|---|---|---|---|
| | + | - | + | - | + | - | + | - | + | - | + | - |
| time | ++ | – | + | – | +++ | — | ++ | – | +++ | - | 39 | 29 |
| psychological comfort | ++ | - | ++ | – | +++ | – | ++ | - | +++ | - | 39 | 13 |
| health | +++ | 0 | +++ | 0 | ++ | 0 | 0 | 0 | + | 0 | 26 | 0 |
| flexibility | ++ | 0 | ++ | 0 | ++ | — | + | – | ++ | 0 | 23 | 13 |
| baggage | + | – | ++ | – | ++ | – | 0 | – | +++ | - | 19 | 23 |
| parking spots | ++ | 0 | ++ | 0 | +++ | 0 | + | 0 | 0 | – | 18 | 8 |
| weather | ++ | — | ++ | – | ++ | – | + | - | + | 0 | 14 | 19 |
| family | ++ | 0 | ++ | 0 | + | 0 | ++ | 0 | 0 | 0 | 14 | 0 |
| children | 0 | – | ++ | - | ++ | - | + | - | + | - | 13 | 9 |
| finances | ++ | 0 | ++ | 0 | + | - | + | – | ++ | - | 13 | 6 |
| distance | ++ | 0 | + | - | + | 0 | + | 0 | ++ | 0 | 12 | 2 |
| surroundings | + | 0 | ++ | 0 | ++ | 0 | + | 0 | 0 | 0 | 12 | 0 |
| social interactions/friends | 0 | 0 | + | 0 | ++ | 0 | + | 0 | + | 0 | 11 | 0 |
| alcohol | 0 | 0 | +++ | 0 | ++ | 0 | + | 0 | 0 | - | 10 | 1 |
| infrastructure | + | – | 0 | - | ++ | — | + | – | + | - | 7 | 24 |
| ecology | + | 0 | + | 0 | 0 | 0 | + | 0 | 0 | - | 6 | 2 |
| comfort (external conditions) | 0 | 0 | 0 | 0 | 0 | 0 | ++ | – | + | 0 | 5 | 2 |
| traffic | + | – | 0 | - | + | – | 0 | 0 | 0 | – | 3 | 16 |
| behavior of others | 0 | 0 | 0 | 0 | + | — | + | – | 0 | 0 | 2 | 13 |
| safety | 0 | – | 0 | - | + | – | 0 | 0 | 0 | - | 2 | 9 |

Legend: +++/— mentioned more than 9 times; ++/– mentioned 3–8 times; +/- mentioned 1–2 times; 0 not mentioned.

Among the most prevalent "new" experiences that the participants had, they often recalled rather the negative ones, e.g., finding out that there was no direct public transport connection to where they needed to go, and/or that the traffic and road conditions of the routes they took frequently were in an inconvenient state; finding out the (high) costs of train tickets; unpleasant encounters with other people on public transport (rude drivers, smelly passengers, overcrowding); major train delays, etc. On the other hand, some respondents made use of their social network and friends when arranging transport and appreciated their children learning to take public transport on their own or that they now seemed to spend more time with their family.

With respect to these findings, we wanted to examine whether a change in attitudes (as measured by the questionnaire) towards different modes of transport also occurred, and more importantly, if a behavioral change was present and lasted beyond the scope of the experiment.

*4.2. Mode Use Frequency and Attitudes Towards Different Modes—Data from the Questionnaire*

Regarding the attitudes (attractiveness, comfort rating, time consumption, finances) towards different transportation modes, overall, no substantial changes between the four time points were observed. Judging from the rankings at all time points, car use was perceived as rather comfortable, time-saving, and attractive, especially by the male participants in the study, although both genders admitted that car use had higher costs. On the other hand, public transport was perceived with mixed feelings by both genders, although the costs were rated as rather low. As for cycling, it was perceived as cost-effective and rather fast, but not always comfortable; men tended to like cycling more than women, who also reported cycling less than men (see further below). Finally, walking was perceived both as cost-effective and very attractive, although slower (more time-consuming) than the other options. But these ratings did not differ significantly between the data collection time points.

Therefore, it was interesting to look at the actual use ("days per week usage") of the respective transportation modes. The overall results are presented in Table 5.

**Table 5.** Days per week usage of different transportation modes at different time points (N = 15).

| | Baseline | | 1 Month | | 3 Months | | 1 Year | | ANOVA | |
|---|---|---|---|---|---|---|---|---|---|---|
| | Mean | SD | Mean | SD | Mean | SD | Mean | SD | F | p |
| Car | 4.40 | 1.45 | 2.20 | 1.52 | 2.87 | 1.64 | 2.93 | 1.71 | 6.552 | 0.001 |
| Public transport | 1.87 | 2.00 | 3.40 | 2.38 | 3.00 | 2.36 | 2.87 | 2.30 | 3.816 | 0.034 |
| Cycling | 2.93 | 2.31 | 2.80 | 2.24 | 3.33 | 2.61 | 2.47 | 2.20 | 1.227 | 0.306 |
| Walking | 3.87 | 2.30 | 4.27 | 2.12 | 4.33 | 2.06 | 4.53 | 1.89 | 0.701 | 0.557 |
| TOTAL TRIPS | 13.07 | 2.96 | 12.67 | 2.32 | 13.53 | 3.18 | 12.80 | 2.88 | 0.535 | 0.583 |

At the one-month time point (i.e., right after the four weeks "without a car"), car use did NOT drop to zero times a week, as would be expected if the participants adhered to the research conditions without exceptions. Nevertheless, the drop in car use was considerable—about 2.2 days per week on average. Three months and one year later, statistically, there was no difference from baseline car use, but in absolute values, the car use per week dropped by about 1.5 days on average. Upon further examination, differences can be observed among the families (Table 6).

**Table 6.** Car use at different time points—days per week.

| Family | Gender | Baseline | 1 Month | 3 Months | 1 Year | Final Evaluation to Baseline |
|---|---|---|---|---|---|---|
| 1 | F | 4 | 0 | 1 | 5 | increased (because of moving) |
| 1 | M | 4 | 1 | 3 | 6 | increased (because of moving) |
| 2 | F | 3 | 2 | 2 | 2 | decreased slightly |
| 2 | M | 4 | 3 | 2 | 2 | decreased |
| 3 | F | | 2 | 1 | 1 | practically no change |
| 3 | M | 2 | 2 | 2 | 1 | practically no change |
| 4 | F | 4 | 4 | 2 | 3 | decreased |
| 4 | M | 4 | 4 | 5 | 2 | decreased |
| 5 | M | 3 | 1 | 1 | 1 | decreased |
| 6 | M | 7 | 1 | 1 | 1 | decreased |
| 7 | M | 5 | 1 | 2 | 3 | decreased |
| 7 | F | 5 | 5 | 5 | 6 | practically no change |
| 8 | F | 5 | 2 | 3 | | decreased |
| 9 | F | 6 | 4 | 5 | 4 | decreased slightly |
| 9 | M | 3 | 3 | 5 | 2 | practically no change |
| 10 | M | 5 | 1 | 5 | 4 | practically no change |
| 10 | F | 7 | 1 | 2 | 2 | decreased |

The biggest change was observed among the three single parents in the experiment: two men (families 5 and 6; both dropped their car use to one trip a week from the previous seven and three trips, respectively) and one widowed woman (family 8), who dropped from five to three car trips a week three months after the experiment. Upon the examination of the interviews and travel diaries, families 5 and 6 both wanted to reduce their car trips before the experiment started. They spontaneously mentioned the habit of using a car and taking part in the experiment as a "good impulse" to realize their intentions. During the experiment, they admitted having to use a car as an emergency option (illness, bigger shopping) and they considered the car as an emergency option after the three months as well. However, they both started to use public transport (and bought/intended to buy a quarter-year season ticket), reported feeling more independent without the car three months after the end of the experiment, and intended to continue living without a car "as best as they can". After one year, it seems as if they kept that promise.

As for family 8, the main reason for their "return to the car" was the cold season. The mother talked about her intention to use a bike more often as soon as the temperature rose, as she did with her children during the experiment. She also reported trying to think about whether all her (car) trips were necessary, which she started doing within the experimental period. Unfortunately, despite numerous

attempts, we were not able to contact this participant after one year to confirm whether her resolution was kept.

A positive trend was also seen in family 2 and family 10, both of them expressing during the interviews that they had considered giving up car use prior to the experiment and the will to discontinue car use as soon as the children were able to travel on their own. Similarly to families 5, 6, and 8 mentioned above, they also did not report having to give up any of their activities completely and considered the no-car-use experience as thoroughly positive. In this matter, the story of family 10, which used to use the car five to seven times a week, is especially interesting: The wife had actually dropped her weekly car use from seven times to twice a week three months after the experiment. The husband's car use returned to its pre-experimental level, but in the interview, this was attributed to his change of workplace to a significantly more distant location (45 min by bus, with a complicated timetable, as compared to 15 min by car). During the post-experimental interviews, the wife reflected on the first two weeks of the experiment as being the hardest because of all the planning involved, but then the family reorganized their trips to school and the children's leisure time activities and the children started to use public transport by themselves, which helped. This new timetable and the behavior of the wife and children still hold.

In the other families, the results were rather mixed: either only one of the partners reduced his/her car use in comparison to the pre-experimental period, or they both returned to their previous car use, mentioning the weather (cold season), comfort, and time (including the cost of trains when travelling with the whole family) as the main reasons in the interviews. One year after the experiment was conducted, family 1 had moved further away from the city, but still maintained their jobs there, which caused them to increase their car use rapidly (the reasons they mentioned were time and comfort when compared to buses or trains).

Regarding the other transportation modes, an increase in public transport use was detected (which more or less prevailed one year after the experiment) and there was no change in cycling or walking. On the family level, the changes usually consisted of adding one trip a week by public transport by one (adult) family member, if any. Practically no change during the whole experiment was observed by two of the families: the one that lived the furthest away (20 km) from the city (where the husband's job was already within walking distance and the wife's job included a lot of travelling between the cities) and one who frequently travelled with two younger children and a lot of stuff. Additionally, one respondent (family 5) only cut down on the total number of trips to the city, making one additional trip by bike. Otherwise, the family members either cut down on their total number of trips as well, or chose a different mode of transportation (walking, cycling, or the car). The biggest change regarding the use of public transport was observed in three families (families 4, 6, and 10), all of them living either in the northwest part of the city or near the main train station. In families 4 and 10, it was mainly the mothers who started out with one trip a week using public transport but increased that to four to seven trips per week even three months after the experiment ended, as did the single parent in family 6. During the interviews, they all explained that they either wanted to "stick with the plan" (and got themselves and the children used to taking public transport), or wanted to preserve the environment and/or be less stressed out about the parking options in the city center.

As for bike use, the average number of trips per week was either already quite high at the beginning of the experiment (five to seven times a week) and did not drop, or it was really low (two trips max.) and did not increase, usually "because of the weather". In three families (families 2, 6, and 8), nonetheless, bike use increased from one or two trips a week to three and as many as seven trips a week. Yet even in these families, the other positive ratings of bike use (attractiveness, comfort, time consumption, finances) did not change significantly, in some cases (finances, attractiveness) probably as a result of the ceiling effect (there were already high ratings at the beginning).

Regarding walking as a transportation mode, in this case, the ceiling effect was more prominent, with three of the families already reporting seven trips per week (e.g., daily use of walking) at the beginning of the experiment (families 3, 7, and 8), and this number either remained the same or

dropped a little if other transportation modes were found (e.g., cycling). A greater increase in walking three months after the experiment ended was observed in three mothers (families 2, 4, and 9), two of them having a newborn baby at the beginning of the experiment (so it probably has to do with the baby's outings) and the last one cutting down on cycling in the winter. The other ratings again remained roughly the same, mostly either because of the ceiling (attractiveness, finances) or floor effect (time consumption).

## 5. Discussion and Conclusions

Our study as a whole shows some interesting findings. Firstly, breaking a habit seems to be difficult, especially for people who are not intrinsically motivated to do so in the first place and who are not using some alternatives already. All of our participants reported struggling with planning ahead, and almost all of them admitted using a car at least once during the experimental period, justifying it with "extraordinary circumstances". As for those respondents who managed to change their habits and continued living with reduced car use even three months and one year after the experiment had ended, we noticed that it was mostly those who were already thinking about reducing their car use before they signed up for the experiment. Additionally, these respondents did not report having to give up any activities during the month without a car and their overall appraisal of the experience was more positive than the evaluation by the other participants, which was rather balanced between positive and negative (e.g., reflecting that although the experience was interesting, living without a car was "hard"). Therefore, we conclude that external incentives might be just the right impulse for people who are on the verge of the decision to reduce their car use and still contemplating it, but intrinsic motivation is necessary to actually change one's habits. Also, a more detailed analysis of the usual routes travelled by the participants—both the ones who changed their habits and the ones who did not—might bring further insights into the relationship between infrastructure and travel mode choice within cities, but the presentation of such an analysis would exceed the scope and length of this article and we therefore decided not to include it.

In the context of motivation and travel mode choice, it is interesting to observe the relationships between car use, attitudes towards different transportation modes, and the reasons/motives people give for their actual behavior. During the initial interview, only two respondents mentioned spontaneously that their car use is rather habitual; a more frequent reason given for car use was "comfort" and "time saving", or "flexibility", which prevailed up until the end of the experiment. These factors even seem to be more important than the admitted financial benefit of travelling via other means of transport (walking, cycling), but this finding might be influenced by the overall financial situation of the respondents, who were mostly university graduates with average or slightly above-average incomes. This may be a limitation of the study, as the level of education can be a contributory factor both for the mode choice and willingness to reduce one's car use. We might assume that possessing a university education is more connected to a healthy and sustainable lifestyle; however, the generalizability of our results is limited. On the other hand, the aim of this study was to repeat the original study by Burwitz, Koch, and Krämer-Badoni [25], where the participants (seven families) were also predominantly university-educated, so in this respect, the replication was successful.

Looking at other factors influencing mode choice, the often-perceived health benefits of cycling or walking, or the relief that one does not have to worry about finding a parking spot, mostly failed to outweigh the perceived comfort of using a car. In light of these findings, appealing only to the financial or health benefits (or the ecological ones, for that matter) of not using a car, when trying to convince people to reduce their car use, might not be effective. Rather, the relevant stakeholders and authorities should focus on providing positive experiences with other modes of transport, mediating, above all, the feeling of comfort and time saving.

This can be done, at least in part, by improving the infrastructure to the needs of non-car-users—as Conley and McLaren [1] or Zavitsas et al. [27] suggest. Most of our participants complained about inadequate public transport or bus/train connections to the places they visit often (i.e., relatives in other

cities or more remote workplaces), longer waiting times, and delays that discouraged them from using public transport, or insufficient infrastructure for cyclists, which made them reconsider using a bike if they had to travel with children. We suggest that improving these aspects might help providing positive experiences, which, in turn, might strengthen the forming of new travelling habits [3,4,34].

As for the role that attitudes play in the decision of (not) using a certain mode of transport, the theory of planned behavior proposes that behavior can be explained by behavioral intention, which in turn is influenced by, among other things, attitudes towards the behavior [31,32]. On the evidence of our results, this assumption might be difficult to prove, as people seem to have difficulty in assessing their attitudes explicitly. When they were asked for their mode choice reasons during the interview, an attitudinal aspect was noticeable in the answers of our participants: They used the words "comfort", "time saving", and "flexibility" to distinguish between the different transportation modes, which can be seen as "rationalizations" of their (un)willingness to use a certain mode of transport, i.e., as an implicit attitude towards the mode of transport. Yet in the questionnaires, when asked explicitly to assess their attitudes, the participants did not change their rankings of attractiveness, comfort, time consumption, and financial costs of different transportation modes across the data collection points, even though some of them did change their actual behavior quite dramatically. One reason for this might be that many people are not used to thinking about their mode choice regularly. It is a habit, as already stated. The questionnaires make them reflect about their attitudes and changes to those. It is possible, thus, that people are not yet fully aware of how they really feel about using different transportation modes and how to formulate their thoughts and feelings. Thus, if they are asked to evaluate their attitudes, the resulting ratings might be biased.

Also, the items in our questionnaire did not take the relative importance of different aspects of travel mode choice (attractiveness, comfort, time consumption, finances) into account, as suggested earlier. For example, although walking was rated as highly attractive, quite comfortable, and very economical, the time consumption rather put the participants off. On the other hand, car use was generally rated as the least economical option, yet when compared to public transport, the perceived comfort benefits usually outweighed this aspect (In this matter, one also has to distinguish between "intra-city" public transport and "inter-city" trains, where the trains were described as rather costly, but this was not reflected in the questionnaire rating of public transport). This might be seen as one of the limitations of our study.

Regarding other limitations of the study, the times of the data collection and the timing of the experimental period itself might have influenced the results. As the experiment took place in autumn/winter, when the weather is usually cold, rainy, or snowy, the willingness to use a bicycle or walk (or even wait for public transport) might have been lower than it would have been had the experiment taken place in spring/summer. For future research, we therefore suggest replicating the experiment in warmer months and observing the seasonal changes in transportation mode use, e.g., every three months for a year. This will enable the effects of the experiment to be distinguished from the seasonal effects more clearly.

Similarly, although we did maintain regular contact with the participants and reminded them of the importance of adhering to the experimental conditions, some participants did not follow the instruction to use at least 10 words for each diary entry either. This certainly influenced both the data quality as well as the experimental effect, precisely as Burwitz, Koch, and Krämer-Badoni [25] already suggested in the frame of their study. Also, most of the participants did not adhere strictly to the "no car use at all" instruction, which in turn might have diminished the intended effect of the experiment. Some bias might be attributed to the different times of the post-test interviews in each family, which allowed them to start using a car again for a few days before the interview was actually conducted. But even from the travel diaries, we could see that the families reported using a car on occasion even during the experimental period. These factors should be taken into account in further replications of such experimental studies.

As for the characteristics of our sample in general, they were surprisingly similar to those of the participants in the study of Burwitz, Koch, and Krämer-Badoni [25]—the people were from different parts of the city, usually around 35 years old (25–50), mostly married, and with a university degree and at least one child. But the fact that they all (or almost all) lived in the same city might represent a limitation of the study; as every city has a unique infrastructure that might influence the traffic mode choice of its citizens in the first place, thus an "across-city" comparison is not easily possible. Also, they were all volunteers, provided with a financial incentive to take part in the experiment. This—together with the above-mentioned university education level—might mean that they were already thinking about their travel behavior and traffic mode change prior to the experiment (which some actually admitted doing), and therefore they were more willing—or "mentally prepared"—to adopt a change in their behavior. On the other hand, the financial incentive might have attracted people who did not care that much about actually changing their behavior but saw the experiment mainly as an interesting opportunity to earn money. We tried to avoid this pitfall by a multiple-stage selection process (people were first contacted before the holidays, and not many of them responded once more in September), but it might still represent an issue with regard to the quality of the data (mostly in the travel diaries), as discussed earlier.

Let us now return to the initial question. Can extrinsically induced experience with "no car use" change future mode choice behavior? The answer, on the evidence of our exploratory study, is yes. Experience with the use of other transportation modes is probably efficient in this respect. If we make people deviate from their usual, habitual transportation mode, they might re-direct their attention to the mode choice process and start re-considering different factors prior to each trip (up until they form a new habit). However, a preliminary set-up for the change is an important pre-condition. There should be other, easier, and cheaper ways to provide incentives to make people try out life, or some time in their life, without a car or with reduced car use. However, facilitating (situational) conditions are important preconditions for a mode shift. Experiencing many problems when trying out "new" travel modes could lead to a boomerang effect: "I tried it and it was really bad; I will never do that again".

As for future research, we plan to work further on this topic to explore and describe, in a narrative way (in-depth interviews), stories of people who do not use a car and how they perceive their mobility preconditions. We will focus on a deeper understanding of life from the perspective of people who never started to use a car, the trigger mechanisms leading to thinking about and actually giving up car use, and changes in the life of those who were forced to give up car use.

A final remark: this study is mostly a qualitative one, for three reasons. First, it is difficult to get the resources (time and money) for, and to carry out, a project like ours on a larger scale. Second, it would be risky to carry out a project on a larger (representative?) scale without earlier experience—which we could make and report on the basis of our study. Third, one of the main topics of our research is motives and we do not see how one can learn to understand the motives of people with the help of standardized, quantitative, or representative procedures, unless there is good preparation for that in the sense of the mixed-methods approach of Cresswell and Plano Clark [46]. We consider our study as one preparatory step in this direction, following the work of Burwitz, Koch, and Krämer-Badoni in 1992 [25]. More research in this direction will follow at the University of Olomouc.

**Author Contributions:** All authors conceived and designed analysis, wrote the manuscript and reviewed the final manuscript. V.L. collected data and performed the analysis.

**Funding:** This work was supported by financial support for university research provided by the Ministry of Education of the Czech Republic, project No. FF_2019_016.

**Conflicts of Interest:** None of the authors are in the conflict of interest.

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
