# Peer review of "Can an Experience with No Car Use Change Future Mode Choice Behavior?"

_sustainability, doi:10.3390/su11174698_

Round 1

Reviewer 1 Report

The paper is an interesting one. 

The experiment is interesting and well performed.
There is a curiosity on the characters of the usual routes generally drove by the volunteers and subsituted by public transport. It could be interesting to understande the average distance travelled and the types of route the volunteers were involved in

Author Response

We understand the curiosity and actually have the data for such an analysis, but as they are quite rich themselves (all the participants kept up daily logs of all of their routes for 30+ days), we concluded that a comprehensive presentation of this data would further extend the length of the article (which is quite long as i tis now). We are considering writing a separate article dealing with this questions. In the current article, we added a short notion about this issue into the Discussion and conclusions section.

Reviewer 2 Report

Topic is interesting and important in times of global warming. Study is mostly descriptive even if mixed methods are used. A sample is quite small but typical in a qualitative research. However, deeper consideration about the sample and possible problems would be good.

I am bit worrying about participant's background. Almost all have university level education. There is a need for short discussion how it affects to results.

Statistics about car use and driving licenses are quite old. Is it really so that there are not newer information about them? The same problem is among earlier studies on car dependency referred in the manuscripts. Most of them are published before the year 2012. Same way it even seems to me that authors have written something about the topic then and now they have used the same references. I can be totally wrong but I am sure there are newer research about the car dependency.

Author Response

Comment:
A sample is quite small but typical in a qualitative research. However, deeper consideration about the sample and possible problems would be good.

Answer:

We agree with this comment. A consideration about sample size characteristics and how it may affect the results was added to the Discussion and conclusions section.

Comment:
I am bit worrying about participant's background. Almost all have university level education. There is a need for short discussion how it affects to results.

Answer:

We agree with this comment. A limitation note was added to the Discussion and conclusions section.

Comment:
Statistics about car use and driving licenses are quite old. Is it really so that there are not newer information about them?

Answer:

We agree with the reveiwer that the statistics are quite old. But there have been no significant changes during the last years. Our point is that for the argumentation that significant measures to reduce car use are needed it is enough to look at the dimension of car use and not so much on the exact figures. The dimension did not change dirung years now.

Comment:

The same problem is among earlier studies on car dependency referred in the manuscripts. Most of them are published before the year 2012.

Answer:

The reviewer is fully correct concerning the studies on car dependency. There are only few of them, one reason being, as quoted in our article (ref. Mayring 2007) that projects dealing with understanding of motives are hardly ever financed. At the same time, much is talked about car dependency, also at congresses, but during our work our impression got stronger that people (colleagues) in this respect talk about what they think (which of course is most probably correct), but not so much about research evidence, in line with what was just said: namely that literature is scarce.

Comment:

Same way it even seems to me that authors have written something about the topic then and now they have used the same references. I can be totally wrong but I am sure there are newer research about the car dependency.

Answer:

Even here our argument from above is valid. Especially work that focusses on behaviour change based on the understanding of motives and on addressing citizens is hardly ever financed. This is also the reason why so long time passed between the study of Burwitz et al. and our study. We consider it essential to take up this string again.

But of course, we have tried to find some more supportive references and also to adapt the text. Because we agree with the reviewer that it is essential that this part of the article makes it crystal clear why our project and why many other projects of the same type are needed.

Round 2

Reviewer 2 Report

The manuscript is now better and acceptable for publishing.

This manuscript is a resubmission of an earlier submission. The following is a list of the peer review reports and author responses from that submission.

Round 1

Reviewer 1 Report

Dear authors,

The study presents an experiment where families are trying to manage everyday life without a car for one month, in an urban area. The topic is relevant and has the potential to bring new insights into the research areas of modal shift and sustainable travel. The authors do not however manage to clearly present the study’s scientific contribution, incl. description of the problem, the aim, the theoretical approach, methods and results of analysis. The structure of the manuscript needs to be improved which includes how the method, data and analysis is presented. Furthermore, the language, grammar and tense needs to be thoroughly checked by an English native speaker.

-Introduction: The problem that constitutes the aim of the study must be highlighted. What is the problem that motivates the authors to explore if families can manage without a car? Why do people have to manage without a car? Environment? Crowded cities? Inactive lifestyles? This needs to be problematized and explained. Further, have there been previous attempts with solving the problem, by previous research? If so, how? What is lacking in previous research that motivates this study?

The aim and research questions should be presented in the introduction.

The introduction lacks ground in previous research, especially since they are so few references in this section. The are some quotes (line 33 and 45) that the reader do not understand where they come from. Who says that? What evidence is there that psychology has always been the answer?

Line 53-45: The authors state that “these questions can only be dealt with in an explorative and qualitative way”. Explain why.

I suggest a section that present previous research relevant to your study.

-Theoretical considerations. The study’s theoretical point of departure needs to be clarified. And how do the theoretical approaches permeate the methods chosen an analysis? Also motivate why your choice of theories is relevant for the study.

2.1 automobility: This section is more of a description of car use and dependency. I suggest that it is moved to a section that presents previous research about the problem. Furthermore, explain what the contribution is with your manuscript concerning this field, i.e. car use.

2.2 Psychological aspects: Besides habit theory, this paragraph actually presents mobility throughout the life course and many social aspects of travel behaviour. As does the results later on. I suggest you consider using the work of Elisabeth Shove and the Social Practice Theory (Shove et al 2012) and research on mobility biographies (Lanzendorf, 2003; Schoenduwe et al. 2015; Müggenburg et al. 2015). As the results are presented now, theories of habit cannot alone explain why the families do as they do.

The theory of habit is insufficiently presented. For example, you state that (line 148) “A habit is developed and then behaviour starts to be performed without further reflection”. I would say that a habit is not developed before a behaviour shows.

Several theories are mentioned (line 166-170). Describe how those are relevant for the study, and if they are not of any specific relevance for your analysis, I suggest you move them to a background-section.

Line 191-197: your hypotheses and expectations are described. Clarity that these are your hypotheses and expectations, I suggest in the introduction.

-Research questions. I suggest that the research questions are reformed. My suggestions:

“1. What advantages and disadvantages are perceived (or experienced) during one month without a car?

2. To what extent can a moth without a car influence travel behaviour change?

A further aim is to communicate positive experiences to authorities and stakeholders. “

Note: aim and research questions should be presented in the introduction section.

-Methodology: Why was families and not individuals chosen as main participants?

It is not clear whether the participants were allowed to get rides from others or use taxi´s.

Reconsider using both table 1 and table 2. Is it possible to merge them to make is easier for the reader? In what way is the children’s ages relevant? I would say that it is relevant, especially in the mobility biographies line of thinking. In what way is education relevant. I suggest you discuss this more clearly in the discussion section. The families’ behaviours are a result of their life phase situations.

Line 252. Please explain “a within-group experimental design”.

Line 254: the items in the questionnaire, where they based on a certain instrument. The validity of the questions need to be discussed.

How are the travel diaries and interviews connected to your theoretical approach? Please explain.

3.4 Methods, this heading is misplaced. All of chapter 3 (Methodology) is actually methods. Consider “data sampling” or data sampling procedure” (or else)

Line 268-280. Consider giving a short summary of the questionnaire instead of presenting all the questions. E.g: The questionnaire contained questions about the use and subjective attractiveness of different transport modes.

Travel diary: it is not clear whether the respondents wrote a diary also during the experiment, or just before and after. Again, consider giving a short summary of the content of the diary instead of writing all questions. The same with the interviews.

-Analysis: 324- 326. The participants reported more/less trips in the journals than in the diaries. The credibility in all data sampling methods can be problematic. I think you should discuss the pros and cons with the chosen data sampling methods. There is no perfect method.

A more thorough description of how the analysis of the diaries and interviews was done must be given. How was the answers subclassified and categorised? What categories did you find? And on what theoretical ground was the narratives of the respondent’s categorized? For example, how is a change in job situation categorised?

-Results. The results are a description of the data, not an analysis of the data. The analysis described in section 3.5 is not shown in the result section. Where are the categories? In order for the reader to understand what your findings are, the analysis needs to be presented, not only the content in the data.

The respondents are referred to as “the three single parents” or “two men” followed by how many children they have. It makes the reading difficult. Instead, refer to the participants a number as they are given in table 1 (and 2). You already explain in the tables if they are male or female and how many children they have. For example, Respondent #1…. Etc.

Table 4: I suggest that you make text out of the table instead. Write how many families that decreased, slightly decreased… etc.  if it is necessary. Note that the reader must be made aware of what slightly and practically no change  and so on means.

In some cases, only the mother or the father in the families took part. It must be made clear how many individuals that took part. Is it relevant to talk about families at all? Consider talking about individuals but refer to their social context.

In the results section, It is not clear where the results come from: questionnaires, interviews or diaries? Clarify where each result come from, perhaps by splitting the results section in different parts. Another suggestion is that you focus this manuscript only on the interviews and travel diaries. It would make sense since you do not compare the different data sampling results anyways.

-Discussion: Line 535-537. You write that “Studies by …//… show that habit affects mode choice intention and behaviour. But it is still unclear what is the strongest factor between rational and habitual behaviour influencing traffic mode choice decision”. Is it only rationality and habits that can explain travel behaviour and modal choice?

Line 538. What are “other factors”?

Line 587: Here, it is clear that not only rationality and habits influence their mode choice decisions. Insufficient public transport is a barrier for mobility in the social environment, which forces people to choose other transport modes. This is an example of how the Social Practice theory by Shove et al. can help to improve your research and analysis. The presence of public transport is as material resource for sustainable mobility.

Line 611: It is not only a qualitative study as you have made quantitative analyses.

Reviewer 2 Report

Introduction-to highlight the reasons for writing the article and the negative consequences of using the car on persons, environment, the society. The authors should show the negative side of the car use habit and why is it necessary to change this behavior and what other studies were used.

I could not find the second direction of the research analyzed separately, although it is a distinct direction. The word sustainable traffic is a keyword but can not be found in the article.

In conclusions, there is no comparison with the results of the research from 1992, how the behaviors have changed in 27 years.

The authors should highlight which are the benefits of the experimental research and for future research what other elements should be taken into account.

Round 2

Reviewer 1 Report

Dear authors,

In the introduction there are still many sayings that are not anchored in the literature, that has no references. One example is “Maybe the problem is greater in the USA and in Canada, but it is probably not that big in most of the cities there. Thus, attempts to make citizens change their mobility behaviour would be worthwhile”.

Also on page 3 there is a lack of references, for example to the critical reports on the car as a transportation mode, in the beginning of page 3, and “It is psychology, stupid” was the answer then”. Who’s answer? Who said that? I understand that you are convinced that psychological theories can explain car use but you have to convince the reader in what way psychology can be used to understand that, with core references in the field. Furthermore, several theories are presented without application on the empirical data. There is no reason to present theories that are not relevant to the scope of your study. Choose a theory if necessary and convince the reader of why that theory is relevant to understand and explain the empirical data in relation to the aim and research questions. Theories of mobility biographies and social practice theory are mentioned in the theoretical section as important theories to undertsndad travel behaviour, but these theories are not used to understand the data.

I suggest that you look at some core references in the field of travel behaviour/transport psychology that has presented empirical data to look at the structure of the papers, their arguments for the research, presentation of findings and discussion of data and previous research.

The results section 4.2 is still a description of data, not a description of the results of the analysis. The reader now has to make the analysis him/herself. Consequently, the reader can not follow how the authors can explain the data with attitudes and habits. In section 4.2, several explanations given by the families can not be solely explained by attitudes and habits, for example time, flexibility, feeling safe and health. The authors also describe main situational factors influencing mode choice, factors that can not be interpreted as attitudinal or habitual only.